# *FDXR*-Associated Oculopathy: Congenital Amaurosis and Early-Onset Severe Retinal Dystrophy as Common Presenting Features in a Chinese Population

**DOI:** 10.3390/genes14040952

**Published:** 2023-04-21

**Authors:** Shutong Yi, Yuxi Zheng, Zhen Yi, Yingwei Wang, Yi Jiang, Jiamin Ouyang, Shiqiang Li, Xueshan Xiao, Wenmin Sun, Panfeng Wang, Qingjiong Zhang

**Affiliations:** State Key Laboratory of Ophthalmology, Zhongshan Ophthalmic Center, Sun Yat-sen University, Guangdong Provincial Key Laboratory of Ophthalmology and Visual Science, 54 Xianlie Road, Guangzhou 510060, China

**Keywords:** *FDXR*, retinal dystrophy 3 congenital amaurosis

## Abstract

Variants in *FDXR* reportedly cause autosomal recessive auditory neuropathy and optic atrophy, expanding to retinal dystrophy. This study aimed to further clarify associated phenotypes. *FDXR* variants were selected from our in-house whole-exome sequencing dataset of 6397 families with different eye conditions. The clinical data of the identified patients were summarized. Biallelic pathogenic or likely pathogenic *FDXR* variants were identified in 11 unrelated patients, including 14 missense variants of which 10 were novel. Fundus observation showed complete optic disc pallor, silver wiring or severe attenuation of retinal vessels, and varying degrees of generalized retinal degeneration. Before the detection of *FDXR* variants, four patients were clinically diagnosed as congenital amaurosis due to the presence of nystagmus a few months after birth, while seven were diagnosed as early-onset severe retinal dystrophy due to the presence of nyctalopia and/or poor vision in early childhood. Biallelic *FDXR* variants are a frequent cause of congenital or early-onset severe retinal dystrophy, especially for patients with severe optic atrophy and retinal dystrophy in early childhood.

## 1. Introduction

Ferredoxin Reductase (*FDXR*) (OMIM 103270) is a mitochondrial protein-coding gene located on chromosome 17q25.1 and first reported by Paul et al. [1]. The encoded flavoprotein is located on the inner mitochondrial membrane and facilitates the synthesis of iron–sulfur clusters and the transport of electrons from NADPH to cytochrome P450 by interacting with ferredoxin 1 and ferredoxin 2. *FDXR* is expressed in all tissues that have mitochondrial P450 systems [2] and plays essential roles in the mitochondrial respiratory chain, steroid biosynthesis, and iron homeostasis [3,4].

*FDXR* was reported to participate in the pathogenesis of autosomal recessive optic atrophy and auditory neuropathy (ANOA, OMIM 617717), which was identified as the top 10 most frequently mutated genes in hereditary optic neuropathies among 2186 probands [1,5,6]. In fundus manifestation, except for the isolated optic neuropathy that is similar to Leber hereditary optic neuropathy (LHON) presentation, retinal dystrophy was also identified among patients with biallelic variants in *FDXR*, which was different from those patients with isolated LHON. In addition, as a mitochondrial specific gene, attention has also been paid to multiple system involvement. Systemic symptoms—the majority being neurological disorders—typically include ataxia, hypotonia, developmental delay, spasticity, seizures, movement disorder, as well as abnormalities in magnetic resonance imaging, the onset or deterioration of which are often associated with infection or other mitochondrial diseases [7]. The autosomal recessive optic atrophy has been reported as the most common ophthalmic presentation, while retinal dystrophy occurred less frequently [1,7,8,9,10,11,12,13]. The reason why variants in such identical genes could contribute to optic neuropathy with or without retinal dystrophy, accompanying with systemic abnormalities in various degrees, was still unknown. It has been suggested that mitochondrial dysfunction caused by reactive oxygen species, iron accumulation, or environmental triggers could contribute to the phenotypic combination optic atrophy and retinal dystrophy.

In the current study, it was hypothesized that through assessing the pathogenicity of detected *FDXR* variants in our exome sequencing data and detailedly analyzing the clinical data of patients with biallelic pathogenic or likely pathogenic variants in *FDXR*, the initial manifestations of these patients could be identified, thus aiding with diagnosis in clinic. As a result, a total of 11 patients were identified with biallelic variants in *FDXR* and presented with Leber congenital amaurosis (LCA) or early-onset severe retinal dystrophy (EOSRD). Combined with reviewing the published data of *FDXR*, the spectrum of *FDXR* was expanded and clarified. The variant spectrum and additional ocular phenotypes that included congenital severe retinopathy resembling LCA and EOSRD of *FDXR* were also expanded.

## 2. Materials and Methods

### 2.1. Probands and Pedigrees

This study was approved by the Institutional Review Board of Zhongshan Ophthalmic Center (KYNL012) and conducted according to the Declaration of Helsinki. Individuals with different eye conditions were recruited from Pediatric and Genetic Eye Clinic, Zhongshan Ophthalmic Center, Guangzhou, China. Written informed consent was obtained from each participant or their guardian. Phenotypic data, venous blood samples, and genomic DNA were collected and extracted as previously described [14].

### 2.2. Identification of FDXR Variants

Pathogenic or likely pathogenic variants in *FDXR* were selected from our in-house whole-exome sequencing (WES) dataset of 6397 families with different eye conditions as of April 2022, and variants in other genes associated with retinal dystrophy or optic atrophy were excluded as previously described (Appendix A) [14]. Pathogenicity of each variant was assessed by comparison with public databases and bioinformatic evaluation. *FDXR* variants with an allele frequency greater than or equal to 0.01, according to the gnomAD database, were excluded. The pathogenicity of *FDXR* missense variants was evaluated using multiple tools, including Combined Annotation Dependent Depletion (CADD) (https://cadd.gs.washington.edu/ accessed on 29 April 2022), Rare Exome Variant Ensemble Learner (REVEL) (https://sites.google.com/site/revel accessed on 29 April 2022), Sorting Intolerant From Tolerant (SIFT) (http://sift.jcvi.org/www/SIFT_enst_submit.html/ accessed on 29 April 2022), Polymorphism Phenotyping version 2 (PolyPhen-2) (http://genetics.bwh.harvard.edu/pph2/index.shtml accessed on 29 April 2022), Protein Variation Effect Analyzer (PROVEAN) (https://provean.jcvi.org/genome/ accessed on 29 April 2022), and Functional Analysis through Hidden Markov Models—Multiple Kernel Learning (fathmm-MKL) (http://fathmm.biocompute.org.uk/fathmmMKL.htm accessed on 29 April 2022). All variants of *FDXR* identified in our cohort was classified based on the American College of Medical Genetics and Genomics and the Association for Molecular Pathology (ACMG/AMP) criteria. A multispecies sequence alignment of the *FDXR* protein was analyzed using Clustal W and the sequences were retrieved from the UniProt Knowledgebase. *FDXR* variants from previous studies and gnomAD were retrieved for comparative analysis with in-house variants. The selected *FDXR* variants were further confirmed by Sanger sequencing. Co-segregation analysis—a method that determines if the biallelic variants in the proband were from one parent each, which suggests the autosomal recessive transmission of *FDXR*—was conducted in available families.

### 2.3. Characterizing the Phenotypes of Patients with FDXR Variants

The clinical records of patients and their family members with biallelic pathogenic or likely pathogenic *FDXR* variants were reviewed systematically, including age of onset, age of exam, initial optic symptoms, best-corrected visual acuity (BCVA), color fundus photographs, scanning laser ophthalmoscopy (SLO), fundus autofluorescence (FAF), optical coherence tomography (OCT), and electroretinography (ERG). Eight published studies concerning biallelic *FDXR* variants were reviewed and the associated clinical data as well as the reported variants of published patients were summarized [1,5,7,8,9,10,11,13].

## 3. Results

### 3.1. Identification of Pathogenic or Likely Pathogenic Variants in FDXR

Biallelic potential pathogenic variants in *FDXR* were detected in 11 unrelated families (Table 1, Figure 1A,B). In total, 14 missense variants were identified, including 10 novel variants (Table 1 and Appendix A). Variants p.R79C, p.R104C, and p.V314L were the 3 most common and account for 13.6% (3/22), 18.2% (4/22), and 13.6% (3/22) of allele numbers, respectively, among all mutant alleles in patients with biallelic pathogenic or likely pathogenic variants in our cohort. All missense variants were confirmed using Sanger sequencing and co-segregation in families with available family members (Figure 1A). These biallelic variants were not detected in samples from normal controls or those related to other eye conditions, suggesting that these biallelic *FDXR* variants were potentially pathogenic. According to the ACMG/AMP guidelines, 6 (p.R193H, p.V314L, p.G325D, p.L232P, p.R275W, and p.W398C) of these potential pathogenic variants were classified as pathogenic and 8 (p.R115G, p.R155G, p.R306H, p.R79C, p.R104C, p.G134R, p.R228W, and p.Q233R) were classified as likely pathogenic. In addition, residue affected by each of the 14 pathogenic or likely pathogenic variants was well conserved through all 8 species (Appendix A).

### 3.2. Clinical Phenotypes of Eight Families with FDXR Variants

The 11 patients with biallelic pathogenic or likely pathogenic variants in *FDXR* were from 11 unrelated families (Table 1 and Table 2, Figure 1A,B). Of the 11 families, 4 (probands of F1–4) were initially considered to have congenital severe retinopathy resembling LCA according to the onset symptoms of nystagmus within the first few months after birth and had poor visual tracking, sluggish pupillary light reflex, and undetectable ERG records during their first visit in the ophthalmic clinic. Similar fundus phenotypes included complete optic disc pallor, silver wiring of retinal vessels, and generalized salt-and-pepper retinal degeneration as demonstrated by RetCam images (Figure 2A–C).

Patient F1-II:1 was born prematurely at 34^+4^ weeks of pregnancy by normal vaginal delivery to parents who were cousins and was found to have nystagmus and poor visual tracking after birth. She had developmental delay and could not lift her head at one year of age on follow-up communication after the identification of *FDXR* variants. No hearing defect was observed in the audiology test. She had hand, foot, and mouth disease, and passed away due to severe pneumonia at 13 months of age. Her monozygotic twin sister had similar eye conditions but did not perform gene testing; she also passed away at 18 months of age from severe respiratory infections due to hand, foot, and mouth disease.

Patient F2-II:2 was born to healthy parents without known consanguinity after a full-term pregnancy. She was found to have nystagmus after birth, and poor vision at one year of age, as she was found to easily bump into things while walking. She had mild mental retardation and BCVA of light perception at eight years of age. No hearing defect or other neurological symptoms were noticed.

Patient F3-II:1 was born at 39^+4^ weeks of normal pregnancy by normal vaginal delivery, and she had nystagmus after birth. She had a febrile convulsion at nine months, with symptoms of microcephaly, hypertonia, and developmental delay. Poor visual tracking, sluggish pupillary light reflex, esotropia, and hyperopia refractive error were also observed in her ophthalmic examinations at the age of one. She was observed to have ataxia at two years of age, and cerebellar atrophy was detected in her brain magnetic resonance imaging, along with an abnormal electroencephalogram. Her BCVA of the last visit was light perception at the age of three.

Patient F4-II:1 had nystagmus and muscle weakness in the lower extremity during infancy. Her brain magnetic resonance imaging results showed normal except for a thick optic nerve at the age of six.

The probands of F5-II:1 and patient F6-II:1 had the manifestation of EOSRD since they all had the onset of nyctalopia or poor vision during early childhood and typical fundus phenotypes. For patient F5-II:1 and patient F6-II:1, they both had the onset of nyctalopia during early childhood and fundus phenotype of optic atrophy, silver-wiring retinal vessels and tapetoretinal degeneration, with the BCVA of 0.1 at 16 years of age in patient F5-II:1 and hand motion (HM) at 19 years old in patient F6-II:1.

Patient F7-II:3 had nyctalopia and strabismus since early childhood but had a relatively good BCVA of 0.3 at 13 and 0.2 at 16 years of age. His fundus presented with optic disc pallor, silver wiring of retinal vessels, and salt-and-pepper retinal degeneration mostly in the mid-peripheral area (Figure 3A,C). FAF indicated hypoautofluorescence outside the vascular arcades and hyperautofluorescent ring in the central macula (Figure 3E). Significant retinal thinning was observed in the OCT image (Figure 3G).

Patient F8-II:1 developed a vision impairment and nystagmus after a fever at the age of five. Complete optic disc pallor, silver wiring of retinal vessels, and generalized tapetoretinal degeneration was observed (Figure 3B,D). Macular involvement was observed in the color fundus photograph, while a hyperautofluorescent ring around the macula and speckled hypoautofluorescence in the mid-peripheral area was observed on wide-field fundus autofluorescence image (Figure 3F). OCT revealed disruptions of the interdigitation and ellipsoid zones, a shallow foveal pit, and remarkably reduced retinal thickness (Figure 3H). Hyperopia was also observed in both patients.

Patient F9-II:1 was found to have nyctalopia at the age of three. Her fundus presented with complete optic disc pallor, silver wiring of retinal vessels, and salt-and-pepper retinal degeneration (Figure 2D). Patients F10-II:1 and F11-II:1 initially experienced poor vision at ages of six and three, respectively. Their posterior fundus presented the complete optic disc pallor, severe attenuation of retinal vessels, and tapetoretinal degeneration (Figure 2E). ERG for both rods and cones were only recorded for patient F11-II:1 in which scotopic and photopic responses were severely reduced (Figure 4).

## 4. Discussion

In this study, a total of 11 unrelated patients were identified with biallelic variants in *FDXR*, in whom 7 probands were observed with EOSRD while the other 4 presented with congenital severe retinopathy resembling LCA. In previous studies, 47 probands have been reported to carry with biallelic pathogenic *FDXR* variants, with autosomal recessive optic atrophy as the main and constant presentation, while retinal dystrophy was less frequently observed [5,10,11,15]. In total, 41 *FDXR* varaints, including 35 missense variants and 6 truncation variants, have been reported in 47 patients from 37 families (Appendix A). Based on reviewing the clinical data of patients in previous studies, ocular defects were mentioned in 45 of the 47 cases associated with biallelic *FDXR* variants. The age of onset of visual defects varied from early infancy to the fourth decade of life, with 31.7% (13/41) in infancy. Nystagmus was found in 18 cases, while autosomal recessive optic atrophy and retinal dystrophy were identified in 42 and 24 cases, respectively (Appendix A). Of these previous reports, images of the fundus were only available for seven individuals from 6 of the 37 families, all demonstrating optic atrophy and vessel attenuation of different severity [5,10,11,13]. Further, retinal pigmentary changes were observed in four of the seven individuals [5]. In our cohort, early-onset severe retinal degeneration, particularly some resembling LCA, was found to be the most common initial phenotype of children carrying biallelic variants in *FDXR*. However, systematic abnormalities were rarely initially identified among our patients. It was valuable for demonstrating the most common ocular phenotypes of *FDXR*, which provided suggestions for recognizing these patients in clinic, especially for an ophthalmologist.

However, slightly different from previous studies concerning *FDXR*, the four patients presenting congenital severe retinopathy resembling LCA and seven cases with EOSRD in our cohort demonstrated more severe phenotypes characterized with complete optic disc pallor, silver wiring, or severe attenuation of retinal vessels, and generalized retinal degeneration. Silver wiring of vessels was also observed in the fundus of patients with variants in *ACO2*, *NMNAT1*, and *IDH3A*, which cause LCA and other retinal dystrophy [16,17,18], which affect the mitochondria, indicating that silver wiring or severe attenuation of vessels may be a common feature in LCA and other retinal dystrophy associated with mitochondrial dysfunction. Based on our large cohort of inherited eye diseases, the LCA and EOSRD phenotype was suggested to be the common presentation of *FDXR* in the Chinese population. However, the *FDXR*-associated phenotype is more heterogenous than assumed. A 25-year-old Chinese woman was referred to an ophthalmology center because of vision loss for over 13 years [13], whose fundus images only demonstrated pale optic discs without any other abnormalities, while in another study, 7 of 10 British patients carrying biallelic variants in *FDXR* manifested with bilateral optic atrophy and retinal dystrophy with or without retinal vessels attenuation. The severity of ocular phenotype might be associated with the pathogenicity of variants, the location of variants, the type of variants, the environmental triggers, or other unidentified factors. More research in more patients with biallelic variants in *FDXR* would be expected to evaluate whether the different phenotypes of *FDXR* were associated with the location of the variants like *CRX* [19], or different stages of the disease progression like *ABCA4* and *CRB1* [20,21]. It has been hypothesized that a cumulative effect rather than an original photoreceptor dysfunction leads to the retinal degeneration in *FDXR*-associated ocular phenotype [22]. As a multifactorial mitochondrial disorder, it is assumed to be affected by poor maternal nutrition or hazardous exposure prior to and during pregnancy, as the bio-energetically impaired mitochondria are likely present in the ovum and transmits to zygote, which act as an additive hit to the mitochondrial dysfunction caused by the varaints in nuclear-encoded mitochondrial genes [22,23]. The potential involvement of epigenetics might be considered [24], but such study on *FDXR* has yet to be explored. While the limitation of our study is that it is mainly based on a local group of people in China, more research on more groups throughout the Chinese population was expected to further confirm our findings.

In this study, four patients were initially considered to have congenital severe retinopathy resembling LCA because they exhibited the major symptoms defining LCA (severe and early visual loss, nystagmus, sluggish pupils, and distinguished ERG records) [25,26]. Fundus changes in patients with LCA were common and varied greatly [14], from relatively normal fundus due to *GUCY2D* variants [27], typical midperipheral changes due to *SPATA7* or *RPE65* variants [28,29], obvious macular atrophy due to *CRB1* variants [21], to severe degeneration involving both of the retina and optic nerve due to variants in *AIPL1* and *NMNAT1* [30,31]. The ocular symptoms and signs in the four unrelated patients with biallelic *FDXR* variants met the criteria for a diagnosis of congenital amaurosis, especially at the early stage when the systemic signs and symptoms have not yet been developed or rarely recognized. Similarly, LCA with extraocular abnormalities has been well documented in systemic diseases, such as Senior-Loken syndrome in which LCA may be noticed in infancy while end-stage renal disease may be observed in teenage years [32]. For *FDXR*-associated congenital amaurosis, the triad fundus changes might be considered as gene-specific, including complete optic disc pallor, silver wiring of retinal vessels, and generalized retinal degeneration. So far, variants in at least 26 genes have been reported to cause LCA [33], in which variants could explain about 60% of the LCA cases. For those LCA cases without identified genetic defects, variants in *FDXR* should be examined, especially for those with the triad fundus changes.

One of the inherited mitochondrial diseases, Friedreich’s ataxia, characterized by dysregulated iron–sulfur cluster biogenesis and iron overload, shares some phenotypes with *FDXR*-associated disorders, including ataxia, sensory neuropathy, and optic atrophy [34]. Identifying the underlying mitochondrial dysfunction mechanism caused by iron–sulfur cluster biogenesis defects, iron-related damage, and potential toxicity will lead to a potential novel strategy for treatment, as some new therapies to treat the severe symptoms of mitochondrial diseases are emerging. Antioxidants (especially idebenone), iron chelators, mitochondrial-related agents, nuclear factor erythroid-derived 2-related factor 2 activators, and gene therapy are potential treatments for Friedreich ataxia [34] and candidates for novel therapies in *FDXR*-associated disorders. A recent study using adeno-associated virus 2 (AAV2) carrying *Fdxr* as a gene transfer vector in a mouse model carrying a variant in *Fdxr* showed significant improvement in optic atrophy, sensory neuropathy, and mitochondrial dysfunction, thus indicating a potential strategy of intervention [4].

The strengths of this study lie in the fact that the variant spectrums of *FDXR* in Chinese were extended and the ocular phenotype spectrums of *FDXR*-associated disease were expanded from previously reported autosomal recessive optic atrophy to congenital amaurosis and early-onset severe retinal dystrophy. The expanding of phenotype spectrum of a such gene with potential fatally systematic abnormalities can help to consolidate clinical diagnosis and potential management in the future. In the ophthalmic clinic, if a child with optic atrophy was identified with biallelic variants in *FDXR*, it is strongly suggested to perform a full-body checkup. In the context of genetic consulting, couples who have already given birth to a patient carrying biallelic variants in *FDXR* are advised to have their embryos or fetus screened for the *FDXR* variants at an early stage. Still, there are some limitations such as lack of mitochondrial-related functional experiment of the patients. In addition, as a mitochondrial disease, how the genetic defects interact with environmental factors and thus trigger the onset of vision loss remained to be elucidated. In conclusion, biallelic pathogenic or likely pathogenic variants in *FDXR* are associated with heterogeneous ocular phenotypes, including EOSRD and congenital severe retinopathy resembling LCA, and similar fundus phenotype was observed in all patients with biallelic pathogenic or likely pathogenic variants in our cohort including complete optic disc pallor, silver wiring or severe attenuation of retinal vessels, and generalized retinal degeneration. Long-term progression of this disease remains a topic for future studies, and further investigations are required to confirm the findings of this study. In addition, more studies are expected in the future to demonstrate the genotype-phenotype correlation of *FDXR* variants.

## Figures and Tables

**Figure 1 genes-14-00952-f001:**
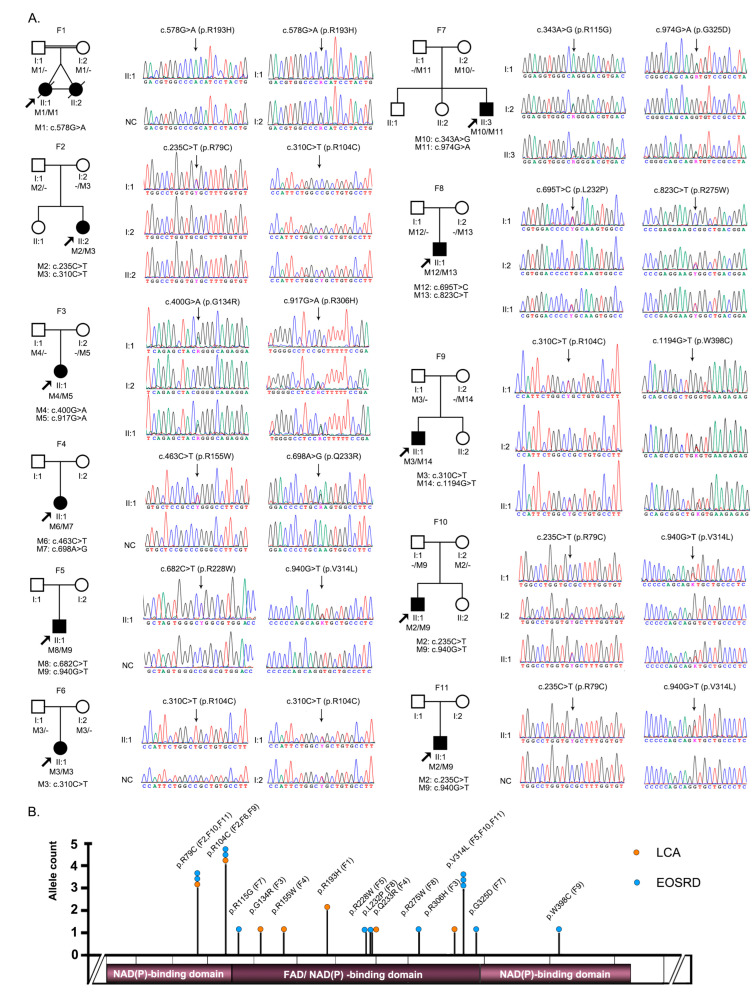
(**A**) Pedigree diagrams and Sanger sequencing of the 11 families with biallelic pathogenic or likely pathogenic *FDXR* variants. (**B**) Schematic diagram of *FDXR* and its variants. EOSRD, early-onset severe retinal dystrophy; LCA, Leber congenital amaurosis; NC, normal control.

**Figure 2 genes-14-00952-f002:**
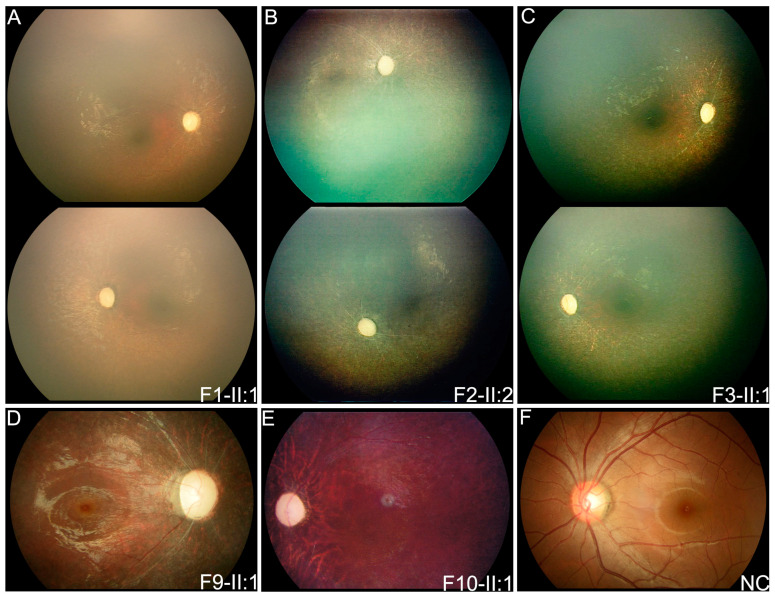
Posterior fundus photographs of three patients with congenital severe retinopathy resembling LCA, two patients with EOSRD, and one normal control. Complete optic disc pallor, silver wiring of retinal vessels, and generalized salt-and-pepper retinal degeneration were observed in RetCam imaging of patient F1-II:1 (**A**), patient F2-II:2 (**B**), patient F3-II:1 (**C**), and in color fundus photograph of patient F9-II:1 (**D**). In patient F10-II:1 (**E**), complete optic disc pallor, severe attenuation of retinal vessels, and tapetoretinal degeneration was observed in color fundus photographs. (**F**) Fundus photograph of a normal control. LCA, Leber congenital amaurosis; EOSRD, early-onset retinal dystrophy; NC, normal control.

**Figure 3 genes-14-00952-f003:**
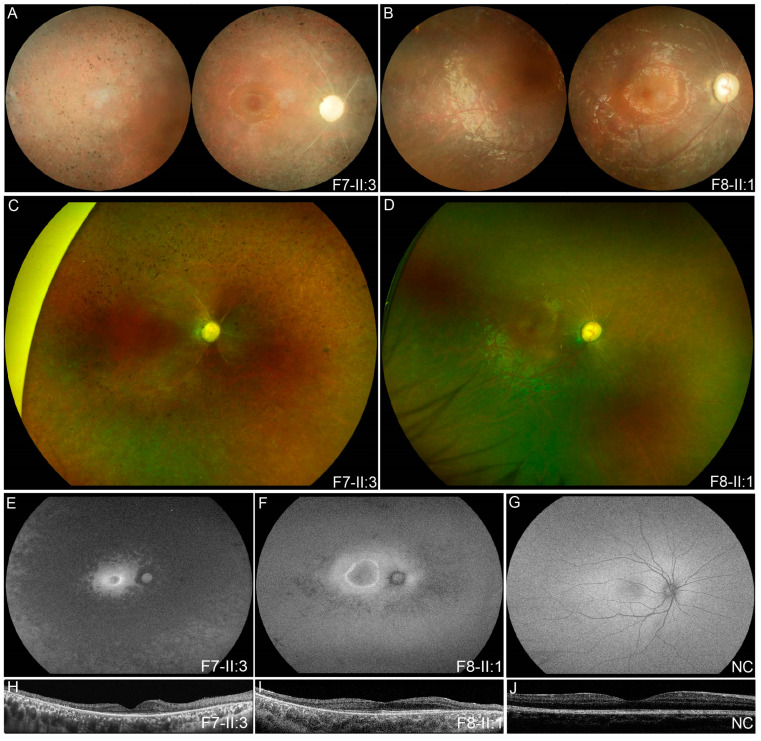
Multimodal fundal images of two probands with EOSRD. Complete optic disc pallor, silver wiring of retinal vessels, and salt-and-pepper retinal degeneration mostly in the mid-peripheral area was observed in color fundus photographs (**A**) and SLO image (**C**) of patient F7-II:3. His FAF image (**E**) indicated hypoautofluorescence outside the vascular arcades and hyperautofluorescent ring in the central macula. Significant retinal thinning was observed in the OCT image (**H**). In patient F8-II:1, complete optic disc pallor, silver wiring of retinal vessels, and generalized tapetoretinal degeneration was observed in his color fundus photographs (**B**) and SLO image (**D**). Macular involvement was observed in the color fundus photograph, while a hyperautofluorescent ring around the macula and speckled hypoautofluorescence in the mid-peripheral area was observed in FAF image (**F**). OCT (**I**) revealed disruptions of the interdigitation and ellipsoid zones, a shallow foveal pit, and remarkably reduced retinal thickness. (**G**,**J**) were FAF image and OCT result of a normal control. EOSRD, early-onset retinal dystrophy; SLO, scanning laser ophthalmoscopy; OCT optical coherence tomography; FAF, fundus autofluorescence; NC, normal control.

**Figure 4 genes-14-00952-f004:**
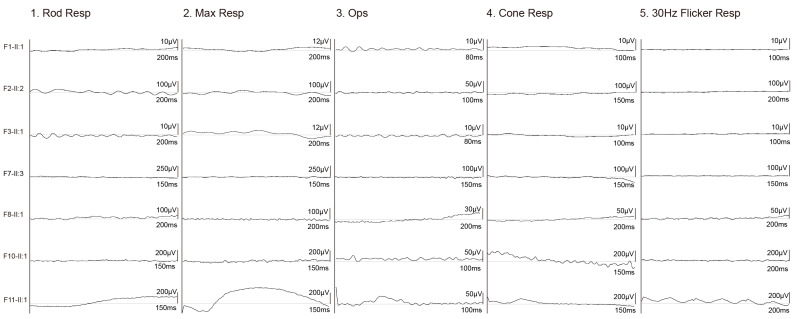
ERG of the seven patients with biallelic *FDXR* variants. ERG for both rods and cones were nonrecordable for F1-II:1, F2-II:2, F3-II:1, F7-II:3, F8-II:1, and F10-II:1, while severely reduced for patient F11-II:1. ERG, electroretinography.

**Table 1 genes-14-00952-t001:** Biallelic pathogenic or likely pathogenic *FDXR* variants in this study.

Variant	Nucleotide	Effect	Bioinformatic Prediction Tools						GnomAD	Previously Reported
ID	Change		REVEL	CADD	SIFT	Polyphen2	PROVEAN	Fathmm-MKL Coding	Frequency	
M1	c.578G > A	p.R193H	0.896	31	DA	PRD	DE	DE	4/248218	Slone J, et al. [6]
M2	c.235C > T	p.R79C	0.653	29	DA	PRD	DE	DE	2/250924	/
M3	c.310C > T	p.R104C	0.517	31	DA	PRD	DE	DE	16/281296	/
M4	c.400G > A	p.G134R	0.818	29.7	DA	PRD	DE	DE	1/249762	/
M5	c.917G > A	p.R306H	0.196	22.8	T	B	DE	DE	2/242172	/
M6	c.463C > T	p.R115G	0.545	24.4	DA	PRD	DE	DE	4/282390	Slone J, et al. [6]
M7	c.698A > G	p.Q233R	0.604	26.5	DA	POD	DE	DE	2/251196	/
M8	c.682C > T	p.R228W	0.793	32	DA	PRD	DE	DE	4/251200	/
M9	c.940G > T	p.V314L	0.13	22.5	T	B	N	DE	18/274502	Song, S, et al. [11]
M10	c.343A > G	p.R115G	0.409	23.6	T	B	DE	DE	/	/
M11	c.974G > A	p.G325D	0.213	24.1	DA	B	DE	DE	/	/
M12	c.695T > C	p.L232P	0.776	29.8	DA	POD	DE	DE	/	/
M13	c.823C > T	p.R275W	0.336	24.3	DA	PRD	DE	DE	2/221422	Jurkute N, et al. [9]
M14	c.1194G > T	p.W398C	0.533	31	DA	PRD	DE	DE	/	/

REVEL = Rare Exome Variant Ensemble Learner, CADD = Combined Annotation Dependent Depletion, SIFT = Sorting Intolerant from Tolerant, PolyPhen-2 = Polymorphism Phenotyping version 2, PROVEAN = Protein Variation Effect Analyzer, fathmm-MKL = Functional Analysis through Hidden Markov Models-Multiple Kernel Learning, DA = Damaging, T = Tolerated, PRD = Probably damaging, B = Benign, POD = Possibly damaging, DE = Deleterious, N = Neutral, DC = Disease causing.

**Table 2 genes-14-00952-t002:** Clinical information of the eight patients with biallelic FDXR variants.

Family	Patient	Nucleotide Change	Gender	Age (Years) At		First	BCVA		Fundus Changes	Others
ID	ID			Onset	Exam	Symptom	Right	Left		
F1	F1-II:1	c.[578G > A]; [578G > A]	F	FMB	0.8	Nystagmus	PVT	PVT	ODP, SRV, SPR	DD
F2	F2-II:2	c.[235C > T]; [310C > T]	F	FMB	2	Nystagmus	PVT	PVT	ODP, SRV, SPR	MR
F3	F3-II:1	c.[400G > A]; [c.917G > A]	F	FMB	1	Nystagmus	PVT	PVT	ODP, SRV, SPR	FC, DD, MW
F4	F4-II:1	c.[463C > T]; [698A > G]	F	FMB	6	Nystagmus	0.05	LP	ODP, SRV, SPR	MW
F5	F5-II:1	c.[682C > T]; [c.940G > T]	M	EC	16	Nyctalopia	0.1	0.025	ODP, SRV, TP	/
F6	F6-II:1	c.[310C > T]; [310C > T]	M	EC	19	Nyctalopia	HM	HM	ODP, SRV, TP	/
F7	F7-II:3	c.[343A > G]; [974G > A]	M	EC	12	Nyctalopia	0.30	0.30	ODP, SRV, SPR	/
F8	F8-II:1	c.[695T > C]; [823C > T]	M	5	6	Poor vision	0.03	LP	ODP, SRV, TP	/
F9	F9-II:1	c.[310C > T]; [1194G > T]	F	3	4	Nyctalopia	0.2	0.25	ODP, SRV, SPR	/
F10	F10-II:1	c.[235C > T]; [940G > T]	M	EC	7	Poor vision	HM	0.04	ODP, SARV, TP	/
F11	F11-II:1	c.[235C > T]; [940G > T]	F	3	5	Poor vision	0.08	0.06	ODP, SARV, TP	/

F = female; M = male; FMB = first few months after birth; EC = early childhood; BCVA = best corrected visual acuity; PVT = poor visual tracking; LP = light perception; HM = hand motion; ODP = optic disc pallor; SARV = severe attenuation of retinal vessels; SRV = silver-wiring retinal vessels; SPR = salt and pepper retinopathy; TP = tapetoretinal degeneration; DD = developmental delay; MR = mental retardation; MW = muscle weakness; FC = febrile convulsion.

## Data Availability

Publicly available datasets were analyzed in this study. This data can be found here: Combined Annotation Dependent Depletion (CADD) (https://cadd.gs.washington.edu/ accessed on 29 April 2022), Rare Exome Variant Ensemble Learner (REVEL) (https://sites.google.com/site/revel accessed on 29 April 2022), Sorting Intolerant From Tolerant (SIFT) (http://sift.jcvi.org/www/SIFT_enst_submit.html/ accessed on 29 April 2022), Polymorphism Phenotyping version 2 (PolyPhen-2) (http://genetics.bwh.harvard.edu/pph2/index.shtml accessed on 29 April 2022), Protein Variation Effect Analyzer (PROVEAN) (https://provean.jcvi.org/genome/ accessed on 29 April 2022), and Functional Analysis through Hidden Markov Models—Multiple Kernel Learning (fathmm-MKL) (http://fathmm.biocompute.org.uk/fathmmMKL.htm accessed on 29 April 2022), Online Mendelian Inheritance in Man (OMIM) (https://www.omim.org/ accessed on 29 April 2022), The Genome Aggregation Database (gnomAD) (www.gnomad-sg.org/ accessed on 29 April 2022).

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
