# Peer review of "FDXR-Associated Oculopathy: Congenital Amaurosis and Early-Onset Severe Retinal Dystrophy as Common Presenting Features in a Chinese Population"

_genes, 2023, doi:10.3390/genes14040952_

Round 1
Reviewer 1 Report
The study presented by Yi et al. provides a comprehensive phenotypic analysis of 11 patients with FDXR mutations and described their clinical features, including retinal dystrophy, optic atrophy, hearing loss, developmental delay, and other systemic manifestations. The authors of this study propose mutations in FDXR as a driving cause of some types of LCA and EOSRD, that until now could have missed their link to a genetic defect.
The manuscript is well-written, clear, and concise. The methods are appropriate and the results are supported by adequate data. The figures are informative and of good quality. The discussion is relevant and balanced. The references are up-to-date and comprehensive.
However, I have some minor comments and suggestions that I hope will help you improve your manuscript before publication:
· The introduction is fairly short. Authors could easily expand the introductory section including more information from previous reports of FDXR-associated disease to provide some background for readers who may not be familiar with this rare condition.
· When it comes to the qPCR results, it would be interesting to see how the authors contrast their findings with those already reported by others (e.g. Protein Atlas).
· It is not completely clear to the reader what is the purpose behind the qPCR-based gene expression analysis. Since there are multiple RNA-seq datasets already published and available, it would be far more interesting to check the differential expression of FDXR across multiple tissues and cell types in that type of data.
· It would be good to include a line or two in the discussion section highlighting the main strengths and limitations of the study. What are the implications of these findings for clinical practice and future research? What are some unanswered questions or challenges that remain to be addressed?
I have also encountered some other minor points that I believe the authors should address:
1. Title should state “FDXR-associated oculopathy: congenital amaurosis and early-onset severe retinal dystrophy as common presenting features in the Chinese population.”
2. Figure 3. Including representative FAF and OCT from a healthy individual will facilitate the reading and understanding of the figure.
3. Line 213: the text refers to proband F8-II:1 as the one with reduced scotopic and photopic responses, however in the figure we that is proband F11-II:1 instead. Please, correct it.
4. Section 3.3. could be part of the discussion paragraph instead.
I hope these comments will be helpful for you to revise your manuscript. I look forward to seeing your revised version soon.
Reviewer 2 Report
In this manuscript by Yi et al., the authors discuss the ocular phenotypes of individuals affected by FDXR mutations in 11 unrelated families in China. The authors also mention new pathogenic variants identified in this study. Such studies are useful in understanding the disease phenotypes associated with gene tic mutations as well as to know various pathogenic variants. It is also important to present these studies in a clear manner with data-supported conclusions.
The comments on the manuscript are listed below:
Title: The title lists that the phenotypes are common among Chinese, however, the studies were limited to the group publishing this work. Unless multiple other groups have shown these exact similar phenotypes throughout Chinese population, it is difficult to make a common statement for a large population based on the geographically localized information. It would be better to state something that clarifies that the statement applies to a local group of people in China.
Please describe what NC means in Figure 1. Also, make sure all labels for DNA sequence listed are correct. For example, F3 lists II:2 in sequence but the actual label (as shown in pedigree chart) should be II:1.
Lines 61-62: “…..variants in other genes associated with retinal dystrophy or optic atrophy were excluded as previously described.” Although the authors provide a reference, not all readers may have access to the IOVS reference cited here. Since understanding the exclusion criteria is important here, please briefly describe the exclusion criteria used.
Line 77: Please explain ‘segregation analysis’ in 1-2 sentences.
Section 2.4 is not needed. It is not one of the ‘Materials and Methods’ and the sentence in section 2.4 (lines 92-94) are exactly the same as in section 2.3 (lines 84-86). The same references are cited as well.
Description in section 3.1 is confusing. When the authors describe the variants, in addition to mentioning the number of families (11), they should also mention that a total of 22 variants were observed.
The authors mention that 10 of the variants are novel (line 108), but also go on to mention that 14 have been classified by American College of Medical Genetics & Genomics (ACMG). However, if 14 out of 22 have been described, it only leaves 8 novel variants.
Further confusing is that the authors list 14 variants in Table 1, but state that 22 variants were studied in this study.
Please describe abbreviation at the first use in main text, e.g., Hand motion for HM in line 192.
Line 212-214: The authors state that ERG for both rods and cones was only recorded for patient F8-II:1 but then show ERG for multiple individuals. How are the traces shown is the recording was not done? Also, the statement in Lines 216-217 is different that lines 212-213. Lines 212-213 state that “ERG for both rods and cones was only recorded for patient F8-II:1…..” while lines 216-217 state that “ERG for both rods and cones was only recorded 216 for patient F11-II:1…..”. Please clarify.
Line 219: Authors state that 41 mutations in FDXR (with 35 missense, and 6 truncation variants) have been reported, but the Table S1 lists 51 variants.
Figure 5: The authors show error bars for most of the samples but not for the ‘retina’ sample. Please describe how many replicates were used per sample, and what kind of replicates were used (was the same cDNA run three times in separate experiments, or were three separate cDNA preps done from the RNA samples, or was the cDNA run in three times in the same experiment) etc.
Line 286: Please also include what gene the AAV2 was carrying.
Please break down section 3.2 in multiple paragraphs. In the current format, all the information is very difficult to follow.
Author Response
Point 1: The title lists that the phenotypes are common among Chinese, however, the studies were limited to the group publishing this work. Unless multiple other groups have shown these exact similar phenotypes throughout Chinese population, it is difficult to make a common statement for a
large population based on the geographically localized information. It would be better to state something that clarifies that the statement applies to a local group of people in China.
Response 1: We agree with the reviewer and have changed the title as follows” FDXR-associated oculopathy: congenital amaurosis and early-onset severe retinal dystrophy as common presenting features in the Chinese population” based on suggestions of Reviewer 1 as well as yours. In the Discussion section, we stated that “Based on our large cohort of inherited eye diseases, the LCA and EOSRD phenotype was suggested to be common presentation of FDXR. While the limitation of our study that mainly based on a local group of people in China, more research on more groups throughout Chinese population was expected to further confirm our findings” has been added in Discussion section as suggested.
Point 2: Please describe what NC means in Figure 1. Also, make sure all labels for DNA sequence listed are correct. For example, F3 lists II:2 in sequence but the actual label (as shown in pedigree chart) should be II:1.
Response 2: Thanks for the comments. We have added the meaning of NC in the figure legend of Figure 1 and replace Figure 1 with the revised label mentioned.
Point 3: Lines 61-62: “…..variants in other genes associated with retinal dystrophy or optic atrophy were excluded as previously described.” Although the authors provide a reference, not all readers may have access to the IOVS reference cited here. Since understanding the exclusion criteria is important here, please briefly describe the exclusion criteria used.
Response 3: Thanks for the comments. We have uploaded a new supplementary Table 3 with the content of genes associated with retinal dystrophy or optic atrophy that were excluded. And the process of whole-exome sequencing was added in methodology.
Point 4: Line 77: Please explain ‘segregation analysis’ in 1-2 sentences.
Response 4: A explain sentence “a method to determine if the biallelic variants the proband carried are from one parent each which suggest the autosomal recessive transmission of FDXR” has been added after “segregation analysis” in Method section as suggested.
Point 5: Section 2.4 is not needed. It is not one of the ‘Materials and Methods’ and the sentence in section 2.4 (lines 92-94) are exactly the same as in section 2.3 (lines 84-86). The same references are cited as well.
Response 5: We agree with the reviewer and had removed Section 2.4 and moved the results of literature review from the Discussion section.
Point 6: Description in section 3.1 is confusing. When the authors describe the variants, in addition to mentioning the number of families (11), they should also mention that a total of 22 variants were observed. The authors mention that 10 of the variants are novel (line 108), but also go on to mention that 14 have been classified by American College of Medical Genetics & Genomics (ACMG). However, if 14 out of 22 have been described, it only leaves 8 novel variants. Further confusing is that the authors list 14 variants in Table 1, but state that 22 variants were studied in this study.
Response 6: Thanks for the comments. The twenty-two in here means total allele count of the variants, which was presented in Figure 1B. In our study, a total of Fourteen variants were observed as mentioned in the second sentence of section 3.1. Variant c.578G>A (p.R193H) and variant c.310C>T (p.R104C) was in homozygous state in F1-II:1 and F6-II:1, respectively. Variants c.235C>T (p.R79C) was observed in F2-II:2, F10-II:1, F11-II:1; variant c.310C>T (p.R104C) was observed in F2-II:2, F5-II:1, F9-II:1; variant and c.940G>T (p.V314L) was observed in F5-II:1, F10-II:1, F11-II:1. The original sentence “Variants p.R79C, p.R104C and p.V314L are the three most common ones and account for 13.6% (3/22), 18.2% (4/22) and 13.6% (3/22)” has been revised to “Variants p.R79C, p.R104C and p.V314L are the three most common ones and account for 13.6% (3/22), 18.2% (4/22) and 13.6% (3/22) of allele numbers”
Point 7: Please describe abbreviation at the first use in main text, e.g., Hand motion for HM in line 192.
Response 7: Thanks for the suggestion. We have added ‘hand motion’ as description for HM in the main text.
Point 8: Line 212-214: The authors state that ERG for both rods and cones was only recorded for patient F8-II:1 but then show ERG for multiple individuals. How are the traces shown is the recording was not done? Also, the statement in Lines 216-217 is different that lines 212-213. Lines 212- 213 state that “ERG for both rods and cones was only recorded for patient F8-II:1.....” while lines 216-217 state that “ERG for both rods and cones was only recorded 216 for patient F11-II:1.....”. Please clarify.
Response 8: Thank you very much indeed. We have corrected this error, in which patient F8-II:1 has been changed to patient F11-II:1.
Point 9: Line 219: Authors state that 41 mutations in FDXR (with 35 missense, and 6 truncation variants) have been reported, but the Table S1 lists 51 variants.
Response 9: Thanks for the comments. A total of 41 mutations have been reported in previous studies, and in Table S1 we added the ten novel variants reported in this study. The Supplementary Table 1 included all variants both reported and in this study.
Point 10: Figure 5: The authors show error bars for most of the samples but not for the ‘retina’ sample. Please describe how many replicates were used per sample, and what kind MDPI | Reply review report of replicates were used (was the same cDNA run three times in separate experiments, or were three separate cDNA preps done from the RNA samples, or was the cDNA run in three times in the same experiment) etc.
Response 10: Thanks for the comments. This part has been deleted based on comments from all of the three reviewers.
Point 11: Line 286: Please also include what gene the AAV2 was carrying.
Response 11: Thank you for pointing out this. It’s Fdxr cDNA that AAV2 is loaded with when inserted into the mouse model carrying a p.Arg289Gln mutation. We have corrected the text in the revised manuscript.
Point 12: Please break down section 3.2 in multiple paragraphs. In the current format, all the information is very difficult to follow.
Response 12: We thank for you for pointing out that. We have break down section 3.2 in multiple paragraphs.
Reviewer 3 Report
The present study is interesting and overall well-conducted. Nevertheless, there are issues to be addressed:
Authors performed a functional evaluation by assessing the expression of FDXR transcript in different purchased extraocular tissues and ocular tissue from a male donor (Paragraph 2.5 in Materials and Methods; paragraph 3.4 in Results):
The utility of this analysis is not well explained throughout the manuscript.
First of all, in the paragraph 2.5 authors should mention which transcript (e.g. NM_024417.5?) they tested by qRT-PCR, since different transcripts leading to a potential functional protein variant are reported on NCBI (https://www.ncbi.nlm.nih.gov/datasets/tables/genes/?table_type=transcripts&key=337ff14a4e3f738c44bb3ccb5d04c2e9).
Notably, the clarification of the transcript is important also for the precise annotation of the variants (i.e. is the coding of the variants referred to the canonical transcript?)
Moreover, did authors check if their data on extraocular tissues are in line with those reported on databases such as GTEX or Human Protein Atlas?
The ocular tissues come from a patient without ocular diseases, but it would have been interesting to compare these “reference” data with those related to affected patients. On this subject, are authors sure that the intracerebral hemorrhage did not affect the expression of FDXR mRNA?
Finally, authors detected missense variants, which are less likely to influence gene expression with respect to other variant types (e.g. stop-codon, frameshift variants..). Are there predictive data supporting an effect of the 14 missense variants on transcript amount?
Authors should better explain and discuss the utility of the gene expression analysis performed on tissues unrelated to their patients or ocular disorders, otherwise it seems not to add any particularly interesting data to the study and could be removed from the manuscript.
Minor comments:
Paragraph 2.2: did the authors evaluate the pathogenicity according to ACMG criteria? If so, a Supplementary Table with the assigned criteria should be added.
Paragraph 2.4: this paragraph is not useful, since any particular criteria for the literature search has been added. If authors consider important this part, the paragraph should be definitely improved, for instance by adding the key words used in Pubmed.
Author Response
Point 1: Authors performed a functional evaluation by assessing the expression of FDXR transcript in different purchased extraocular tissues and ocular tissue from a male donor (Paragraph 2.5 in Materials and Methods; paragraph 3.4 in Results): The utility of this analysis is not well explained throughout the manuscript. First of all, in the paragraph 2.5 authors should mention which transcript (e.g. NM_024417.5?) they tested by qRTPCR, since different transcripts leading to a potential functional protein variant are reported on NCBI(https://www.ncbi.nlm.nih.gov/datasets/tables/genes/?table_type=transcripts&key=337ff14a4e3f738c44bb3ccb5d 04c2e9). Notably, the clarification of the transcript is important also for the precise annotation of the variants (i.e. is the coding of the variants referred to the canonical transcript?) Moreover, did authors check if their data on extraocular tissues are in line with those reported on databases such as GTEX or Human Protein Atlas? The ocular tissues come from a patient without ocular diseases, but it would have been interesting to compare these “reference” data with those related to affected patients. On this subject, are authors sure that the intracerebral hemorrhage did not affect the expression of FDXR mRNA?. Authors should better explain and discuss the utility of the gene expression analysis performed on tissues unrelated to their patients or ocular disorders, otherwise it seems not to add any particularly interesting data to the study and could be removed from the manuscript.
Response 1: We agree with you and has deleted this part from the manuscript based on comments from all of the three reviewers.
Point 2: Paragraph 2.2: did the authors evaluate the pathogenicity according to ACMG criteria? If so, a Supplementary Table with the assigned criteria should be added.
Response 2: A new Supplementary Table 2 with evaluation of the pathogenicity according to ACMG criteria has been suppled as suggested.
Point 3: Paragraph 2.4: this paragraph is not useful, since any particular criteria for the literature search has been added. If authors consider important this part, the paragraph should be definitely improved, for instance by adding the key words used in Pubmed.
Response 3: Thank you for your suggestions. Based on your suggestion, the paragraph 2.4 has been removed from our manuscript.

Round 2
Reviewer 2 Report
The authors have addressed the comments. Thank you for clarifying the information about variants. Few more minor updates are requested.
1. In the title, please change ‘the Chinese population’ to ‘a Chinese population’.
2. Line 69: In line 55, the authors changed the description from ‘potential pathogenic variants’ to ‘likely pathogenic variants’ but then again use the term ‘potentially pathogenic variant’ in line 69.
3. Please read through the texts to correct typographical and grammatical errors.
